



# Electron heating by HF-pumping of high-latitude ionospheric F-region plasma near magnetic zenith

Thomas B. Leyser[1], Björn Gustavsson[2], Theresa Rexer[2], and Michael T. Rietveld[3]

[1]Swedish Institute of Space Physics, Uppsala, Sweden.
[2]The Arctic University of Norway, Tromsø, Norway.
[3]EISCAT Scientific Association, Ramfjordmoen, Norway.

**Correspondence:** Thomas Leyser (thomas.leyser@irfu.se)

**Abstract.** High frequency electromagnetic pumping of ionospheric F-region plasma at high and mid-latitudes gives the strongest plasma response in magnetic zenith, antiparallel to the geomagnetic field in the northern hemisphere. This has been observed in optical emissions from the pumped plasma turbulence, electron temperature enhancements, filamentary magnetic field-aligned plasma density irregularities, and in self-focusing of the pump beam in magnetic zenith. We present results of EISCAT (European Incoherent SCATter association) Heating-induced magnetic-zenith effects observed with the EISCAT UHF incoherent scatter radar. With Heating transmitting a left-handed circularly polarised pump beam towards magnetic zenith, the UHF radar was scanned in elevation in steps of 1.0° and 1.5° around magnetic zenith. The electron energy equation was integrated to model the electron temperature and associated electron heating rate and optimized to fit the plasma parameter values measured with the radar. The experimental and modeling results are consistent with pump wave propagation in the L mode in magnetic zenith, rather than in the O mode.

## 1 Introduction

A powerful high frequency (HF) electromagnetic wave transmitted from the ground into the ionospheric F-region stimulates the strongest plasma response on long time scales in the direction antiparallel to the geomagnetic field in the northern hemisphere as seen from the HF transmitter. This magnetic zenith effect has been observed in several ways for a range of pump frequencies in experiments at high and mid latitudes.

In experiments with the EISCAT (European Incoherent SCATter association) high power HF facility Heating in Norway in 1999, pump-induced optical emissions were imaged unambigously for the first time (Brändström et al., 1999). Gustavsson et al. (2001) presented tomography-like estimates of the volume distribution of the 630.0-nm emissions from these experiments and found that the emissions intensified and self-focused towards magnetic zenith during the 4-min pumping. Kosch et al. (2000) observed that while the HF beam was directed vertically the region of maximum optical emissions was displaced towards magnetic zenith as seen from EISCAT Heating. The authors also noted that published data of coherent HF radar scatter off geomagnetic field-aligned density irregularities tend to maximise in the magnetic field-aligned direction.

Radio tomography and scintillations using amplitude and phase measurements on the ground of VHF signals from orbiting satellites were used to study HF pump-induced electron density modifications in experiments with the mid-latitude Sura HF





facility in Russia. Small-scale filamentary magnetic field-aligned plasma density irregularities were found to be strongest in magnetic zenith, both when the Sura beam was vertical or at an angle in between the vertical and magnetic zenith (Tereshchenko et al., 2004). Further, initial experiments with the Sura HF beam directed either 12° south of vertical or 16° both showed the strongest optical emissions at 630.0 nm near magnetic zenith at 18–19° south (Grach et al., 2007).

Rietveld et al. (2003) scanned the EISCAT Heating beam between three elevations from vertical to near magnetic zenith
and found the electron temperature enhancements to be almost always the strongest in the magnetic zenith position. When the EISCAT UHF radar was scanned between the same positions the strongest electron heating was always observed near magnetic zenith. In addition, optical emission at 630.0 nm were localized near magnetic zenith and HF coherent radar scatter off geomagnetic field-aligned density striations maximized when the Heating beam was in magnetic zenith. Blagoveshchenskaya et al. (2006) too observed the strongest field-aligned density striations when the Heating beam was in magnetic zenith.

Honary et al. (2011) examined the temporal evolution of the magnetic zenith effect as observed in the electron temperature measured by the EISCAT UHF radar. The beams from the Heating facility and the UHF radar were alternatively directed vertically and in magnetic zenith. Maximum temperature enhancements were observed when both the Heating and radar beams were in magnetic zenith. Further, these electron temperature enhancements reached a stationary state already within 10 s after pump-on in the 60 s on/90 s off pump cycle.

The magnetic zenith effect in optical emissions has also been observed in experiments with the HAARP (High frequency Active Auroral Research Program) facility in Alaska, USA (Pedersen and Carlson, 2001; Pedersen et al., 2003). Further, Pedersen et al. (2008) determined the optical emission production efficiency as a function of angle by HF-beam swinging experiments. The maximum emission efficiency occurred exactly in the geomagnetic field-aligned position.

Kosch et al. (2007) observed self-focusing of the pump beam in magnetic zenith in experiments at HAARP. The pump-
induced optical emissions at 557.7 nm collapsed from a cone of approximately 22° to 9° within tens of seconds after pump-on, while cycling the pump 60 s on/60 s off.

In the present treatment we report experimental results on the magnetic zenith effect obtained with the EISCAT Heating facility. The F-region plasma response to the HF pumping was observed with the EISCAT UHF incoherent scatter radar that was scanned in steps of either 1.0° or 1.5° around magnetic zenith to measure the electron temperature and other plasma
parameter values. Nonlinear least squares analysis was used to fit electron temperature profiles obtained from integrating the electron energy equation with a parametrized heat source, taking into account heat conduction, electron heating and cooling, with measured plasma parameters. The analysis gave the electron heating rate as a function of altitude and elevation angle.

## 2 Experiment setup

The EISCAT Heating experiments were performed during daytime in November 2014 and October 2017. The Heating facility
transmitted a left-handed circularly polarised wave (LHCP, often referred to as O mode) in a beam directed towards magnetic zenith ($\sim 78°$ elevation south) and cycling 150 s on/85 s off. The term "left-handed" is defined with reference to the geomagnetic field direction; the electric field rotates in the opposite sense to the gyromotion of electrons.





Plasma parameter values in the F region were obtained with the EISCAT UHF incoherent scatter radar. The radar measurements utilized the Beata modulation scheme which includes a 32-bit binary alternating code with a baud length of 20 $\mu$s. The

UHF radar beam was scanned in steps of $1.0°$ in the experiment in November 2014 and in steps of $1.5°$ in October 2017, between eight elevations around magnetic zenith in the plane containing the vertical and with a duration of 5 s in each position. The radar beam width was approximately $0.5°$. The pump cycle of 150 s on/85 s off enabled appropriate coverage of the radar measurements throughout the pump-on time, so that after several pump cycles under stable ionospheric conditions the temporal evolution during the pumping could be obtained at all elevations. The radar data analysis provided 5 s temporal resolution and

15–20 km range resolution, depending on range.

## 3 Experimental results

Figure 1 displays measured height profiles for the electron concentration ($\widetilde{N}_e$), electron temperature ($\widetilde{T}_e$) and ion temperature ($\widetilde{T}_i$) for the experiment on 25 November 2014 (the tilde denotes measured parameters as opposed to modelled). For this case the pump frequency $f_0 = 6.30$ MHz, which is approximately half way between the fourth and fifth electron gyroharmonic in

the F region. The transmitted power was 818 kW. For some unknown technical reason, Heating did not transmit a circularly polarised wave during this experiment: the effective radiated power (ERP) was 242 MW in LHCP and 157 MW in right-handed circular polarisation (RHCP), assuming perfectly conducting ground. However, electron heating effects from pumping with LHCP dominate over those with RHCP. Bryers et al. (2013) estimated the height-integrated heating source for O-mode pumping to be approximately a factor of three larger than for X-mode pumping, for a pump duty cycle of 50% and the O-mode

pump frequency not near an electron gyroharmonic, however, with the X-mode frequency near a gyroharmonic (their Figure 5). In addition, the ERP in our experiments for LHCP was larger than for RHCP. We therefore consider the measured heating effects to be representative of pure LHCP pumping.

The $\widetilde{N}_e$ profile in the top panel of Fig. 1 is stable throughout the displayed time interval and does not show modulations due to the pumping. However, the $\widetilde{T}_e$ profile in the middle panel exhibits clear pump-induced modulations. The HF pumping is

marked by white boxes and the red zigzag line indicates the radar elevation scan. The $\widetilde{T}_i$ shown in the bottom panel does only exhibit weak pump-induced modulations. The ionospheric conditions and response to the HF pumping in the experiments on 24 October 2017 were similar to those shown in Fig. 1. However, whereas for 2014 the ionospheric critical frequency $foF2$ was near 8 MHz, well above $f_0$, $foF2$ was near $f_0$ in 2017.

The used pump cycle in combination with the radar scan cycle enabled measurement of the temporal evolution of the

ionospheric parameters at all radar elevation angles throughout the pumping. Figure 2 shows the temporal evolution of height profiles of $\widetilde{T}_e$ for the different elevation angles of the UHF radar, starting at $t = 5$ s after pump-on for the experiments on 25 November 2014. Such measurements require reasonably stable ionospheric conditions during several pump cycles (see Fig. 1), as the radar, scanning eight elevations between $75.2°$ and $82.2°$, samples the interaction region at a given elevation at different times after pump-on in different pump pulses. With measurements at sufficiently many pump pulses the temporal evolution can

then be traced throughout the duration of pump-on ($t = 0$–150 s) at all elevations.



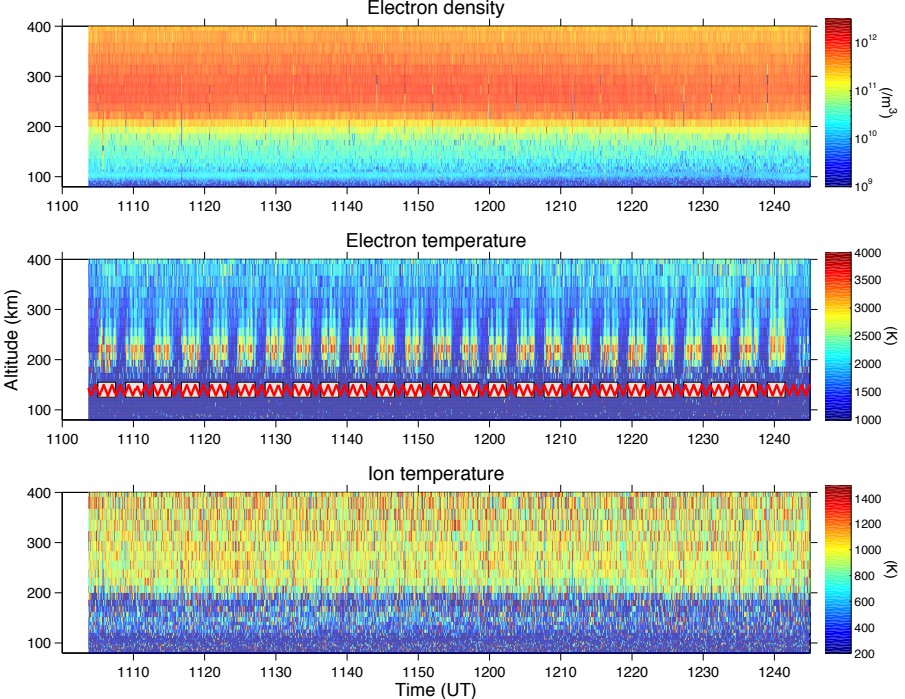

**Figure 1.** Height profiles as a function of time of $\widetilde{N}_{\mathrm{e}}$ (top panel), $\widetilde{T}_{\mathrm{e}}$ (middle panel) and $\widetilde{T}_{\mathrm{i}}$ (bottom panel) during pump cycling on 25 November 2014. The white boxes in the middle panel show pump-on and the red zigzag line indicates the elevation of the UHF radar which was scanned between $75.2°$ and $82.2°$ in $1.0°$-steps.

As seen in Fig. 2, $\widetilde{T}_{\mathrm{e}}$ enhancements occurred already within the first few seconds of pump-on. The high $\widetilde{T}_{\mathrm{e}}$ around 300 km in the first 5-s data dump is likely not real, but due to HF enhanced ion acoustic lines on the topside ionosphere. During the following few tens of seconds $\widetilde{T}_{\mathrm{e}}$ was further enhanced at all elevations and in a wider altitude range. Notice also the slow conduction of electron heat toward increasing altitudes with time, up to 300–400 km altitude, as can be seen in Fig. 1 too. The strongest $\widetilde{T}_{\mathrm{e}}$ enhancements occurred at the elevations $77.2°$ to $79.2°$, around magnetic zenith ($\sim 78°$). This is also where the $\widetilde{T}_{\mathrm{e}}$ enhancements extended toward the highest altitudes. Differences in the enhanced $\widetilde{T}_{\mathrm{e}}$ profiles can be discerned even though the radar elevation changes by only $1.0°$.

Figure 3 displays $\widetilde{T}_{\mathrm{e}}$ height profiles versus time for the experiment on 24 October 2017. As for Fig. 2, the high $\widetilde{T}_{\mathrm{e}}$ around 300 km in the first 5-s data dump is likely not real. In this experiment $f_0 = 6.2$ MHz, which again is approximately half way between the fourth and fifth electron gyroharmonic in the F region. The transmitted power was 734 kW and the ERP was 471 MW (LHCP). The radar was scanned in steps of $1.5°$ from $74.56°$ to $85.06°$, which is a larger range of elevations than that for the $1.0°$-steps in Fig. 2. $\widetilde{T}_{\mathrm{e}}$ enhancements due to the HF pumping again occurred already in the first 5-s radar data integration after pump-on and $\widetilde{T}_{\mathrm{e}}$ was the highest at the elevations $77.56°$ and $79.06°$, closest to magnetic zenith ($\sim 78°$). The gaps in the plots are because the ionospheric conditions were not stable long enough to give sufficient data to obtain the

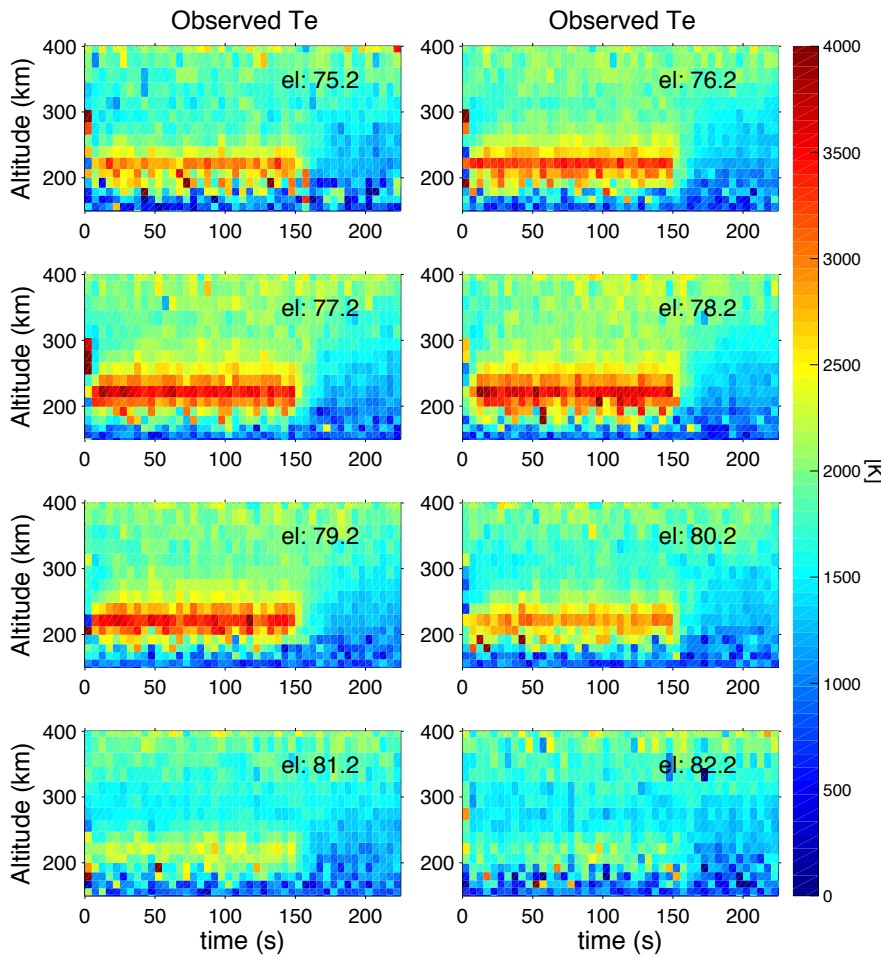

**Figure 2.** Height profiles of $\widetilde{T}_{\mathrm{e}}$ (colour coded) versus time for the different elevation angles of the UHF radar between $75.2°$ and $82.2°$ on 25 November 2014 (11:03:45–13:00:00 UT). Pump-on was from $t = 0$ to $t = 150$ s.

full temporal evolution at all elevations. However, the results for $\widetilde{T}_{\mathrm{e}}$ are similar to those in Fig. 2 for the experiments on 25 November 2014.

## 4   Electron heating model

To get information on the source that underlies the observed electron temperature enhancements we model the electron heating rate through the fluid equations (Shoucri et al., 1984). As the measurements of the UHF incoherent scatter indicate no major

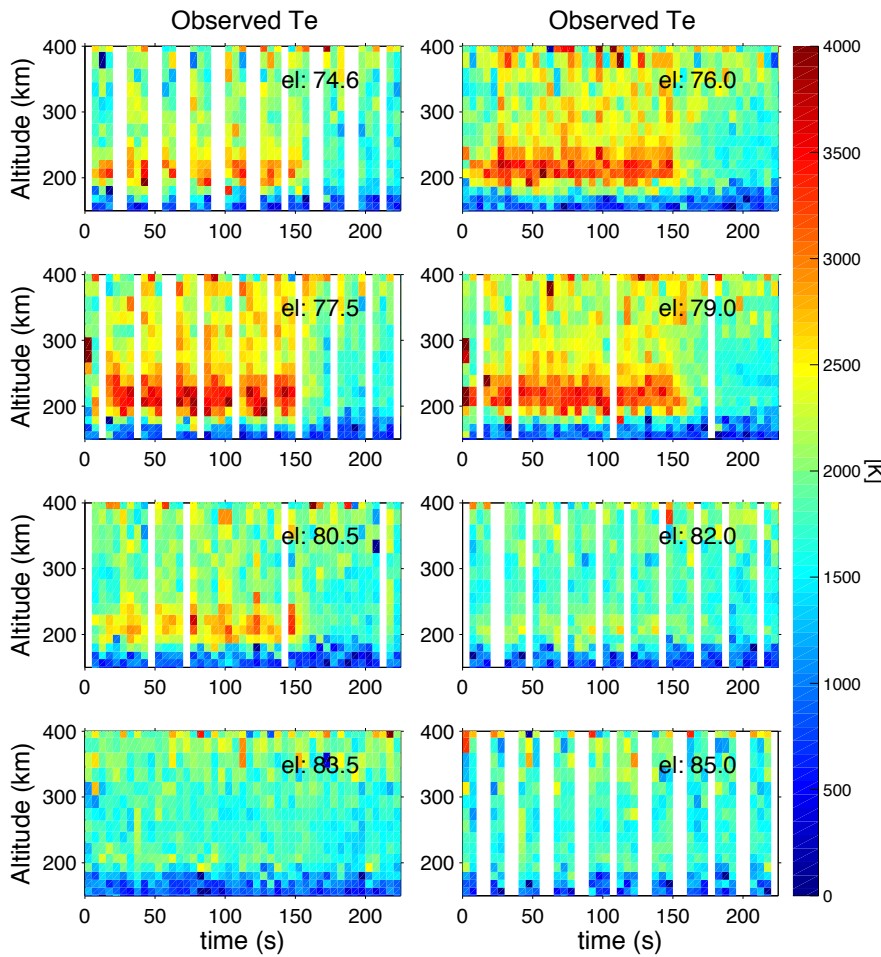

**Figure 3.** Height profiles of $\widetilde{T}_{\mathrm{e}}$ (colour coded) versus time for the different elevation angles of the UHF radar between $74.56°$ and $85.06°$ on 24 October 2017 (12:00:00–12:43:00 UT). Pump-on was from $t = 0$ to $t = 150\,\mathrm{s}$.

pump-induced effects in $\widetilde{N_{\mathrm{e}}}$ and $\widetilde{T}_{\mathrm{i}}$ (Fig. 1), the fluid equations can be reduced to the electron energy equation:

$$\frac{3}{2}N_{\mathrm{e}}k_{\mathrm{B}}\left(\frac{\partial T_{\mathrm{e}}}{\partial t} + (\mathbf{v}_{\mathrm{e}}\cdot\hat{\mathbf{z}})\frac{\partial T_{\mathrm{e}}}{\partial z}\right) + N_{\mathrm{e}}k_{\mathrm{B}}T_{\mathrm{e}}\frac{\partial}{\partial z}(\mathbf{v}_{\mathrm{e}}\cdot\hat{\mathbf{z}}) = \frac{\partial}{\partial z}\left(\kappa_{\mathrm{e}}\frac{\partial T_{\mathrm{e}}}{\partial z}\right) + Q_{\mathrm{HF}} + Q_{\mathrm{e}} - L_{\mathrm{e}} \tag{1}$$

where $T_{\mathrm{e}}(z,t)$ is the modelled electron temperature, $\hat{\mathbf{z}}$ is the unit vector in the direction of the geomagnetic field, $k_{\mathrm{B}}$ is Boltz-mann's constant, $\kappa_{\mathrm{e}}(T_{\mathrm{e}},z,t)$ is the electron heat conductivity, $Q_{\mathrm{HF}}$ is the HF pump wave energy deposition to the electrons, $Q_{\mathrm{e}}(z,t)$ is the background electron heating rate (mainly from photoelectrons), and $L_{\mathrm{e}}(T_{\mathrm{e}},z,t)$ is the electron cooling rate due

to elastic and inelastic collisions with ions and neutrals.





With negligible plasma drift along the geomagnetic field as measured with the UHF radar, the convective terms in Eq. (1) can be neglected, giving (Löfås et al., 2009; Gustavsson et al., 2010):

$$\frac{3}{2}N_e k_B \frac{\partial T_e}{\partial t} = \frac{\partial}{\partial z}\left(\kappa_e \frac{\partial T_e}{\partial z}\right) + Q_{HF} + Q_e - L_e \tag{2}$$

The heating rate of the electrons due the electromagnetic pump wave consists of two parts:

$$Q_{HF} = Q_\Omega + Q_{AA} \tag{3}$$

where $Q_\Omega$ is the ohmic heating due to collisional damping of the pump wave and $Q_{AA}$ is the heating due to the anomalous absorption of the wave associated with the excitation of, for example, upper hybrid turbulence and associated small-scale density striations. The ohmic heating rate is the time averaged product of the pump electric field $\mathbf{E}_0$ and induced electric current $\sigma_{ij}\mathbf{E}_0$, where $\sigma_{ij}$ is the conductivity tensor: $Q_\Omega = (1/2)Re[\mathbf{E}_0^* \cdot (\sigma_{ij}\mathbf{E}_0)]$ (Gustavsson et al., 2010). At the relatively high ERP levels used in the experiments, $Q_{AA}$ gives the dominating contribution to $Q_{HF}$ and may be several times larger than $Q_\Omega$ (Bryers et al., 2013).

In the present treatment we obtain a model $T_e(z,t)$ of the observed $\widetilde{T}_e(z,t)$ by integrating the electron energy equation (2). The electron heating rate $Q_{HF}(z,t)$ due to the HF pumping is modeled by a one-dimensional and asymmetric gaussian along the geomagnetic field. $Q_{HF}(z,t)$ has its maximum $Q_m$ at range $z_0$ and has independent upper ($\sigma_u$) and lower ($\sigma_d$) half-widths (Senior et al., 2012; Bryers et al., 2013):

$$Q_{HF}(z,t) = \begin{cases} Q_m \exp\left[-(z-z_0)^2/\sigma_d^2\right]\{1-\exp\left[-(t-t_{on})/\tau\right]\}, & z < z_0 \\ Q_m \exp\left[-(z-z_0)^2/\sigma_u^2\right]\{1-\exp\left[-(t-t_{on})/\tau\right]\}, & z \geq z_0 \end{cases} \tag{4}$$

where $t_{on} \leq t \leq t_{off}$ is the time during which HF pumping occurs. This leads to a parameter estimation problem in the model parameters $Q_m$, $z_0$, $\sigma_d$, $\sigma_u$, and $\tau$, that we solved by weighted nonlinear least squares:

$$par_{HF} = argmin \sum \left[\frac{\widetilde{T}_e(z,t) - T_e(z,t,par_{HF})}{\sigma_{\widetilde{T}_e}}\right]^2 \tag{5}$$

where $T_e(z,t,par_{HF})$ is obtained by integrating Eq. (2) with $Q(z,t,par_{HF})$ and $\sigma_{\widetilde{T}_e}$ is the standard deviation of the observed electron temperature.

When integrating Eq. (2) we used the observed range profiles for $\widetilde{T}_i$ and $\widetilde{N}_e$ as they evolve in time at each elevation. For example, $L_e$ depends both on $T_i$ and $N_e$ and both the left-hand side of Eq. (2) and $\kappa_e$ depend on $N_e$. As initial condition we took a smoothed $\widetilde{T}_e$ range profile measured just before pump-on. Further, we used mixed boundary conditions, taking at the lower boundary $T_e = \widetilde{T}_e(z = 150\,\text{km}, t)$ as given by the UHF radar measurements at $z = 150$ km slightly before pump-on at $t = t_{on}$ and at the upper boundary $\partial T_e/\partial z(z = 500\,\text{km}, t) = 0$. The fixed temperature at the lower boundary follow from the observations with the additional theoretical justification that at such low altitudes $T_e$ and $T_i$ are both approximately equal to the neutral temperature due to the high collision frequencies. The upper boundary condition too is based in the observations, and corresponds to a balance between upward heat flux out from the ionosphere and downward heat flux from the magnetosphere into the ionosphere.



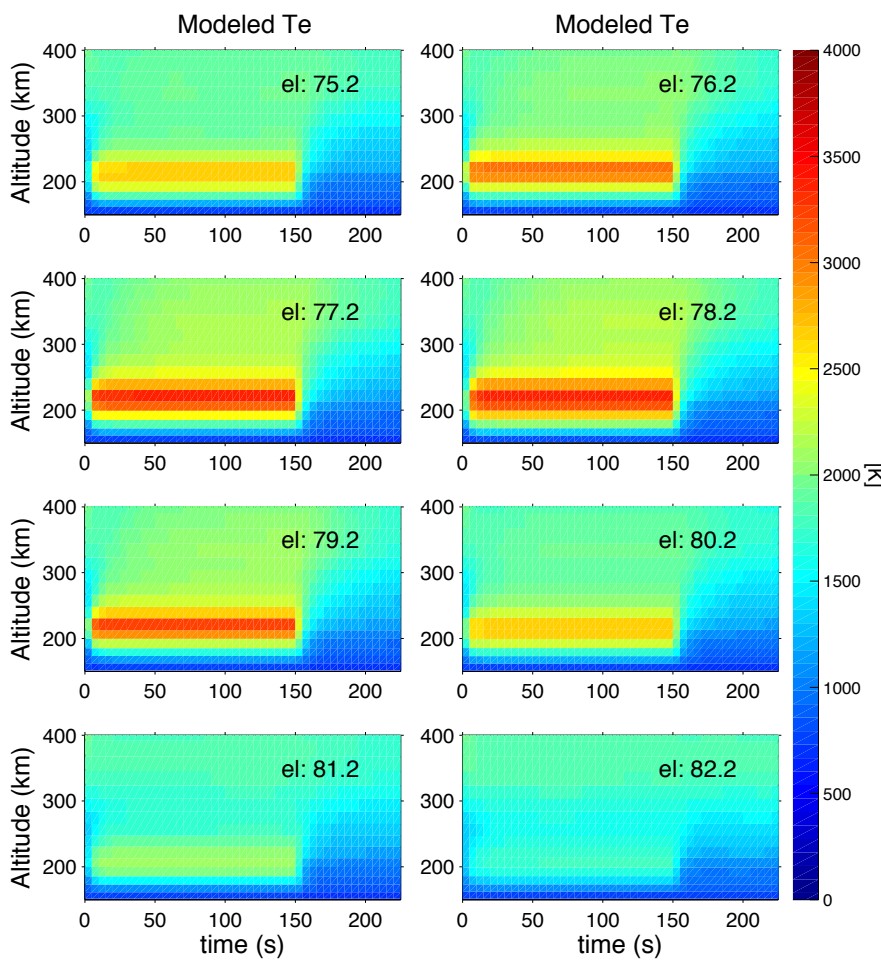

**Figure 4.** Modelled temporal evolution of the altitude profile of the electron temperature $T_e(z,t)$ for the different elevation angles in the experiments on 25 November 2014 (Fig. 2).

## 5 Modeling results

The temporal evolution of the modeled $T_e$ altitude profile for the elevation angles scanned by the radar are obtained by integrating Eq. (2) with the optimal parameters for $Q_{HF}(z,t)$. The results are shown in Figs. 4 and 5, which correspond to the measurements in Figs. 2 and 3, respectively. $T_e(z,t)$ is enhanced for all elevations already within the first seconds after pump-

on at $t = 0$ s. Slow conduction of the electron heat is seen both upward and downward in altitude and $T_e(z,t)$ reaches the highest values near magnetic zenith ($\sim 78°$). The modeling results in Figs. 4 and 5 agree quantitatively with the measurements in Figs. 2 and 3, respectively.





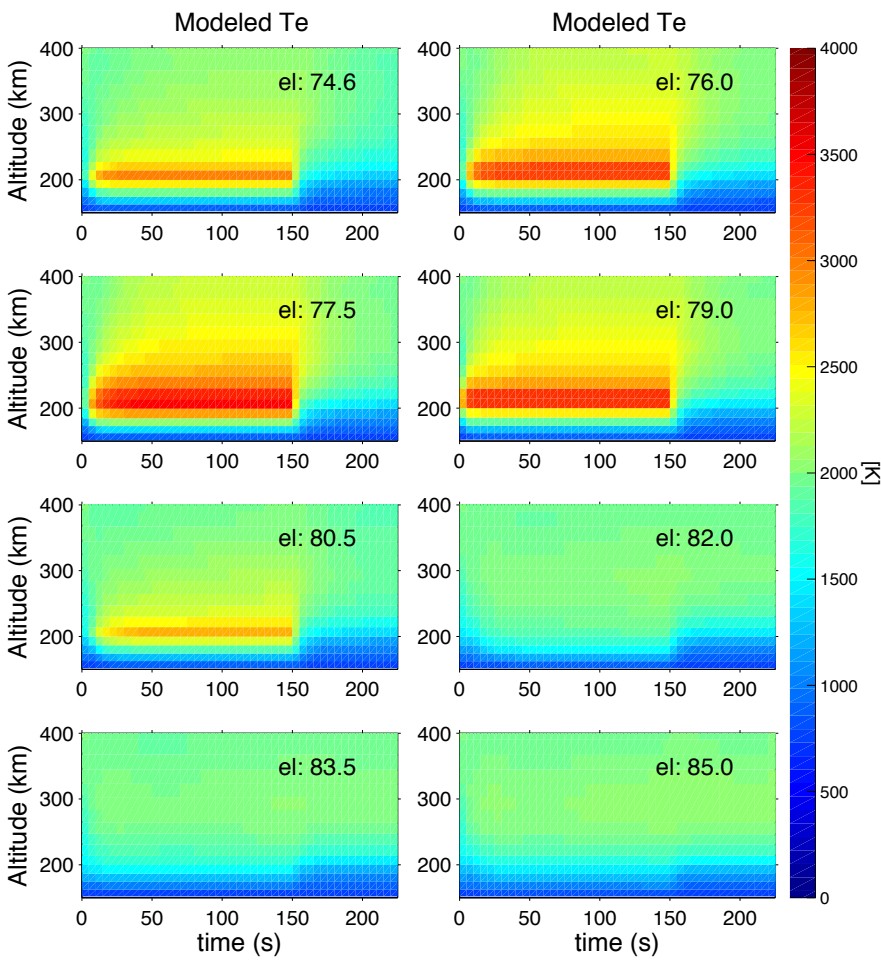

**Figure 5.** Modelled temporal evolution of the altitude profile of the electron temperature $T_e(z,t)$ for the different elevation angles in the experiments on 24 October 2017 (Fig. 3).

Figure 6 displays the corresponding $Q_{HF}$ versus elevation angle for the experiment on 25 November 2014 (Fig. 2) in the two top panels and for 24 October 2017 (Fig. 3) in the two bottom panels. The left panels show the column-integrated $Q_{HF}$

(blue) as well as the profile of the transmitted Heating beam (red) and the right panels depict the altitude profiles of $Q_{HF}$. These modeling results are for the case after that steady state was reached in the 150-s pump-on period. The white and black lines in the right panels show the altitude of the plasma and upper hybrid resonances, respectively, as obtained from the ion and plasma lines. The altitude separation between the two resonances is larger in the lower panel for which $f_0$ was near $foF2$ than in the upper panel for which $f_0$ was well below $foF2$.



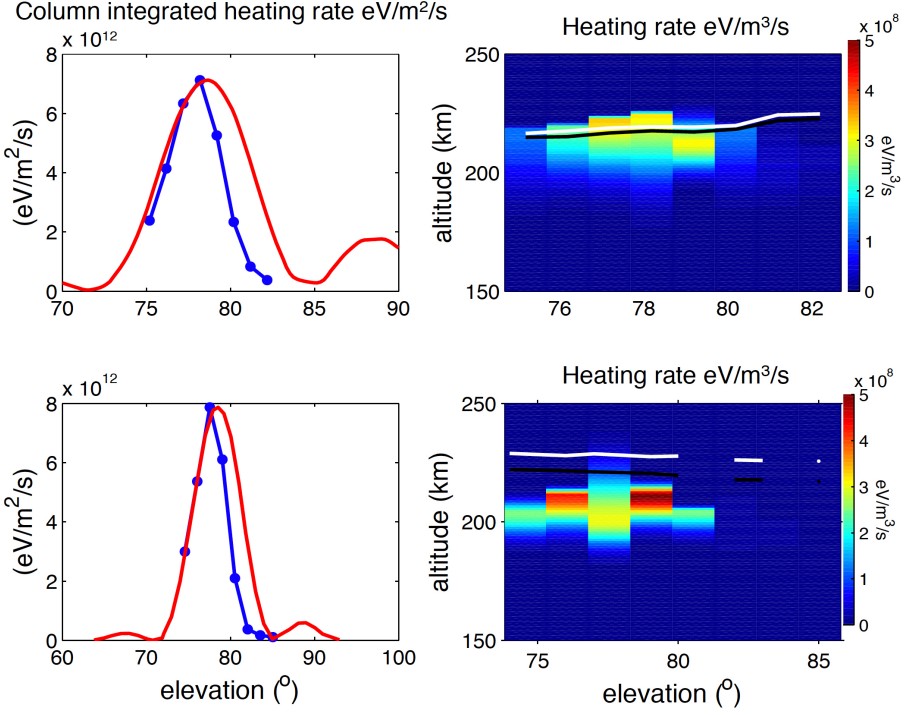

**Figure 6.** Modelled electron heating rate $Q_{HF}$ (eV/m³/s) during steady state versus radar elevation angle for 25 November 2014 (upper panels) and 24 October 2017 (lower panels). The left panels display the column-integrated $Q_{HF}$ (blue dots, with connecting lines to guide the eye) and the relative intensity of the transmitted Heating beam (red) assumed to propagate in vacuum. The right panels show the altitude profiles of $Q_{HF}$. The white line indicates the altitude of the plasma resonance where $f_p = f_0$ and the black line shows the upper hybrid resonance height at which the upper hybrid frequency equals $f_0$. Note that the elevation scale is different in the upper and lower panels.

The column-integrated $Q_{HF}$ in the top panel is maximum at 78.2° and in the bottom panel at 77.5°, which are the same elevations at which the observed electron temperature reached the highest values (Figs. 2 and 3, respectively). Also, the column-integrated $Q_{HF}$ is maximum at the elevation closest to magnetic zenith ($\sim 78°$).

      Further, as seen in the left panels of Fig. 6, the angular extent of the $Q_{HF}$ profile is smaller than that of the Heating beam. $Q_{HF}$ follows the profile of the Heating beam at elevations lower than magnetic zenith ($\sim 78°$) while at higher elevations it is
more confined to magnetic zenith than the Heating beam. For reference, the Spitze angle is about 6° from the vertical or at about 84° elevation.

      As seen in the right panels of Fig. 6, the altitude profile of $Q_{HF}$ is generally asymmetric, with $Q_{HF}$ decreasing steeply with increasing altitude above the maximum and declining more gradually with decreasing altitude below the maximum. Only in the bottom panel for 77.5° (at magnetic zenith), $Q_{HF}$ decreases more slowly toward high altitudes than toward lower altitudes.
Also, $Q_{HF}$ decreases steeply for increasing elevations beyond about 80°, towards the vertical. This decrease of $Q_{HF}$ with





increasing elevation is steeper than what would be expected from the point view of the width of the Heating beam in vacuum (see the left panels).

It is notable that $Q_{\mathrm{HF}}$ reaches larger values at the two elevations next to magnetic zenith compared to for the elevation nearest to magnetic zenith. In the top right panel of Fig 6, $Q_{\mathrm{HF}}$ is slightly higher at $77.2°$ and $79.2°$ than at $78.2°$, while the $Q_{\mathrm{HF}}$ profile is more extended in altitude at $78.2°$. The bottom right panel shows larger differences, with $Q_{\mathrm{HF}}$ higher at $76.0°$ and $79.0°$ than at $77.5°$, while the $Q_{\mathrm{HF}}$ profile is more extended in altitude at $77.5°$. Thus, despite that the maximum of the $Q_{\mathrm{HF}}$ profile is slightly lower at magnetic zenith compared to at the two nearest neighboring elevations, the column-integrated $Q_{\mathrm{HF}}$ is maximum at magnetic zenith (left panels) for both experiments.

## 6  Discussion

We have presented experimental and modelling results concerning electron heating and the ionospheric plasma response to HF pumping near magnetic zenith. The experiments were performed with the EISCAT Heating facility and measurements of the plasma response were done with the EISCAT UHF incoherent scatter radar. The Heating beam was tilted in the magnetic zenith direction and the UHF radar was scanned between eight positions around this direction to study the electron-heating efficiency. The electron heating rate $Q_{\mathrm{HF}}(z,t)$ and associated electron temperature $T_{\mathrm{e}}(z,t)$ due to the HF pumping was modeled by integrating the energy equation (2) and fitting the model parameters with respect to the measurements of $\widetilde{N}_{\mathrm{e}}$, $\widetilde{T}_{\mathrm{e}}$ and $\widetilde{T}_{\mathrm{i}}$.

Differences in the plasma response were observed for radar elevations differing by only $1°$ (Fig. 2). The pump-induced measured $\widetilde{T}_{\mathrm{e}}(z,t)$ enhancements (Figs. 2 and 3), the modelled $T_{\mathrm{e}}(z,t)$ (Figs 4 and 5) and associated column-integrated $Q_{\mathrm{HF}}$ (Fig. 6) were all found to maximise in the magnetic zenith direction ($\sim 78°$ elevation). Further, the angular width of the $Q_{\mathrm{HF}}$ profile, with a full width at half maximum (FWHM) of about $4°$ around magnetic zenith, was less than that of the HF beam, which suggests that some focusing of the Heating beam occurred.

Pedersen et al. (2008) obtained the angular distribution of the optical emission production efficiency by HF beam swinging experiments at HAARP. The optical emission production efficiency peaked at magnetic zenith with a FWHM of $7°$, for which the HAARP beam width and many other experiment variables were accounted for. This FWHM of $7°$ is larger than the two cases for $Q_{\mathrm{HF}}$ in Fig. 6. The HAARP experiments used an ERP of 32.1 MW at 2.83 MHz and 42.4 MW at 3.3 MHz, thus, both lower ERP and lower $f_0$ than in the present EISCAT experiments. It is plausible that self-focusing effects were larger at the higher ERP in the present experiments which could give a more narrow region of pump-induced enhancements.

It has been proposed that filamentary plasma density ducts can guide a transmitted LHCP wave, entering the ionosphere in the O mode, as an L-mode wave along the geomagnetic field (Leyser and Nordblad, 2009; Nordblad and Leyser, 2010). The L mode is an LHCP electromagnetic wave mode with the wave vector parallel or anti-parallel to the ambient magnetic field. For a homogeneous and cold magnetized plasma the refractive index ($n_{\parallel}$) parallel to the ambient magnetic field is given by $n_{\parallel}^2 = 1 - f_{\mathrm{p}}^2 / f(f + f_{\mathrm{e}})$, where $f_{\mathrm{p}}$ is the electron plasma frequency and $f_{\mathrm{e}}$ is the electron gyrofrequency. Whereas the O mode has a cutoff at $f_{\mathrm{p}} = f_0$, the L mode has the cutoff frequency $f_{\mathrm{L}} = -f_{\mathrm{e}}/2 + (f_{\mathrm{p}}^2 + f_{\mathrm{e}}^2/4)^{1/2}$ which corresponds to $f_{\mathrm{p}} \approx f_0 + f_{\mathrm{e}}/2$



for $f_\mathrm{p}^2 \gg f_\mathrm{e}^2$ for a pump wave at the frequency $f = f_0$. Thus, an electromagnetic wave in the L mode can propagate at higher plasma densities than in the O mode.

L-mode propagation can occur when the background plasma density gradient near the plasma resonance is parallel to the geomagnetic field, instead of the density gradient for example being vertical as in a horizontally stratified ionosphere. Such a condition with the density gradient being magnetic field-aligned can occur in density ducts, either natural or pump-induced. In the L mode the pump wave can propagate upwards passing through the plasma resonance on its way to the cutoff at $f_\mathrm{p} \approx f_0 + f_\mathrm{e}/2$ if the plasma is sufficiently dense. With its perpendicular electric field, strong pumping of upper hybrid phenomena

localized in small-scale density striations and related anomalous electron heating can occur at higher altitudes and deeper into the plasma compared to the case of an O-mode wave which therefore could contribute the strong plasma response observed in magnetic zenith (Leyser and Nordblad, 2009; Nordblad and Leyser, 2010).

The electron heating rate $Q_\mathrm{HF}$ due to the HF pumping obtained for our experiments was found to exhibit an interesting dependence on the elevation angle near magnetic zenith (Figs. 6). $Q_\mathrm{HF}$ is maximum at the elevations next to magnetic zenith.

At magnetic zenith, the $Q_\mathrm{HF}$ profile is more extended in altitude, such that the column-integrated $Q_\mathrm{HF}$ is maximum in this direction. These results are consistent with the pump wave propagating in the L mode in magnetic zenith and in the O mode at angles deviating from the zenith direction. Evidence of L-mode propagation of the EISCAT Heating beam has previously been obtained as transionospheric propagation for $f_0 < foF2 < f_0 + f_\mathrm{e}/2$, in which case an L-mode wave would not be reflected but pass through the ionospheric plasma density peak, by direct measurement on the CASSIOPE spacecraft (Leyser et al.,

2018) and indirectly by EISCAT UHF radar observations of ion acoustic lines in the topside ionosphere (Rexer et al., 2018).

Figure 6 also displays the altitude of the plasma resonance (white line in the right panels). The position of the $Q_\mathrm{HF}$ profile relative to the plasma resonance is not fully understood. In the top right panel, $Q_\mathrm{HF}$ is maximum slightly above the plasma resonance at magnetic zenith. This is consistent with that a pump wave in the L mode can propagate well above the plasma resonance, whereas an O-mode wave cannot. Further, an L-mode wave has its electric field perpendicular to the geomagnetic

field all the way up to its reflection height, so that pumping of upper hybrid turbulence can occur in an extended altitude range. In the O mode, on the other hand, the electric field turns to parallel to the geomagnetic field close to the reflection height which favours excitation of Langmuir turbulence that generally causes less electron heating than upper hybrid turbulence.

However, in the bottom right panel of Fig. 6 all electron heating appears to occur well below even the upper hybrid resonance height (black lines). In this case $foF2$ was near $f_0$ whereas for the top panel it was well above $f_0$, which is consistent with

that the altitude separation between the plasma and upper hybrid resonances is larger in the bottom panel. We do not have any explanation for why electron heating seemed to occur at such low altitudes in this case.

Gurevich et al. (2002) developed a theory for self-focusing of the electromagnetic pump wave propagating in the O mode on geomagnetic field-aligned density striations. An important mechanism in the nonlinear pump beam self-focusing is the trapping of pump rays near the magnetic zenith direction in the large-scale density depletions within the beam, as previously

was found in numerical studies (Gurevich et al., 1999). The results were shown to be consistent with observations of pump-induced optical emissions at HAARP (Pedersen et al., 2003). However, the possibility of propagation of the pump wave in the L mode, deeper into the plasma than what is possible in the O mode, was not considered. Whereas the nonlinear self-focusing



of the pump beam is an important mechanism, particularly for guiding the pump beam in magnetic zenith, it does as it stands
not seem to account for the difference that we have found in the altitude distribution of the electron heating rate in magnetic
zenith compared to that just about 1° away from this direction (Fig. 6). We therefore suggest that such theories for self-focusing
are developed to include the possibility of L-mode propagation.

## 7    Conclusions

The EISCAT Heating facility was used to pump ionospheric F-region plasma by cycling 150 s on/85 s off with an LHCP HF
beam directed in magnetic zenith. Plasma parameter values were measured with the EISCAT UHF incoherent scatter radar that
was scanned in steps of 1.0° and 1.5° in elevation around magnetic zenith. The temporal evolution of the electron temperature
profile was modelled by integrating the electron energy equation, which was used to fit the measured plasma parameter values
with a model electron heating rate.

The observed electron temperature enhancements and the associated column-integrated electron heating rate and modelled
electron temperature all exhibit maxima in magnetic zenith. In addition, the altitude range of electron heating is more extended
in magnetic zenith than for elevations deviating from the zenith direction. These results are consistent with pump wave propa-
gation in the L mode rather than purely O mode and suggest the importance of L-mode propagation for understanding magnetic
zenith effects.

*Data availability.*  Access to the raw data may be provided upon reasonable request to the authors.

*Author contributions.*  Björn Gustavsson, Thomas Leyser and Michael Rietveld performed the experiments. Gustavsson developed the theo-
retical model and made the numerical analysis. Theresa Rexer contributed to the data analysis and data presentation. Leyser carried out most
of the interpretation and prepared the paper. All the co-authors helped in the interpretation of the results, read the paper and commented on
it.

*Competing interests.*  The authors declare that they have no conflict of interest.

*Acknowledgements.*  EISCAT is an international association supported by research organisations in China (CRIRP), Finland (SA), Japan
(NIPR and ISEE), Norway (NFR), Sweden (VR), and the United Kingdom (UKRI)





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
