# Peer review of "Electron heating by HF-pumping of high-latitude ionospheric F-region plasma near magnetic zenith"

_Annales Geophysicae, 2019_

## Referee Comment (RC1) · Anonymous Referee #1 · 10 Jan 2020

This is a well written and clear manuscript, introducing the idea that L-mode radio wave propagation may play an important role in ionospheric modification experiments and in the magnetic zenith effect in particular. In the past, only O-mode radio wave propagation was considered. I can recommend this manuscript for publication. I have only two relatively minor comments:

L56: The authors should include the Heater beam width please. The UHF radar beam width is explicitly mentioned, but the Heater beam width is only discovered by the reader when they get to Figure 6.

L216-217: This sentence regarding L-mode versus O-mode radio wave propagation

requires a little more explanation or at least a reference please. Otherwise, the reader is forced to simply accept the authors' words.

---

## Referee Comment (RC2) · Anonymous Referee #2 · 27 Jan 2020

This study examines two intervals where electrons in the high-latitude ionosphere were heated using the EISCAT high-frequency heating facility. Plasma parameters were then measured using the EISCAT incoherent scatter radar. The results were combined with modelling to show that the L-mode wave propagation leads to the plasma properties measured at magnetic zenith, as opposed to O-mode wave propagation as was previously thought.

This manuscript is very well written and concise. I believe that the results are new and of interest to the scientific community and I therefore recommend this paper for publication, subject to some minor modifications (see below).

[Figure]

Minor comments: 1) L. 151: Instead of "quantitatively", I think you mean "qualitatively". Whilst the colours and patterns in the data look similar, the numbers do not exactly match. 2) Your use of hyphenating O-mode/O mode is inconsistent. Please choose one. 3) Either in the introduction or the experimental setup section: Please reference L- and O-mode and explain the significance of this experiment. Whilst I understand that this paper shows a new result, it does not explain very well why this is a) important and b) useful to know. 4) Figure 6: Please indicate on the plots where the zenith angle is (perhaps add a dashed line). This would help the reader a lot to understand your argument.

---

## Author Comment (AC1) · 31 Jan 2020

We thank referee #1 for constructive comments on the manuscript. Our reply to the items raised be the referee is the following:

L56: Done on line 58.

L216-217: An explanation of L-mode versus O-mode propagation has been added in lines 220-227.

---

## Author Comment (AC2) · 31 Jan 2020

We thank referee #2 for constructive comments on the manuscript. Our reply to the items raised be the referee is the following:

1) Done (line 154).

2) "O-mode" (and "X-mode" and "L-mode") is hyphenated when it comes before a noun that is modified (so called compound adjective), otherwise not.

3) A sentence mentioning the findings concerning L mode and O mode has been added at the end of the Introduction (lines 53-54).

[Figure]

4) The elevation corresponding to magnetic zenith has been marked on all four plots in figure 6 and is also described in the figure caption.